# Rectifying Shortcut Behaviors in Preference-based Reward Learning

**Wenqian Ye**
University of Virginia
wenqian@virginia.edu

**Guangtao Zheng**[*]
Accenture
zhguangt@gmail.com

**Aidong Zhang**
University of Virginia
aidong@virginia.edu

## Abstract

In *reinforcement learning from human feedback*, preference-based reward models play a central role in aligning large language models to human-aligned behavior. However, recent studies show that these models are prone to *reward hacking* and often fail to generalize well due to *over-optimization*. They achieve high reward scores by exploiting shortcuts, that is, exploiting spurious features (e.g., response verbosity, agreeable tone, or sycophancy) that correlate with human preference labels in the training data rather than genuinely reflecting the intended objectives. In this paper, instead of probing these issues one at a time, we take a broader view of the reward hacking problem as *shortcut behaviors* and introduce a principled yet flexible approach to mitigate shortcut behaviors in preference-based reward learning. Inspired by the invariant theory in the kernel perspective, we propose Preference-based Reward Invariance for Shortcut Mitigation (PRISM), which learns group-invariant kernels with feature maps in a closed-form learning objective. Experimental results in several benchmarks show that our method consistently improves the accuracy of the reward model on diverse out-of-distribution tasks and reduces the dependency on shortcuts in downstream policy models, establishing a robust framework for preference-based alignment.

## 1 Introduction

Reinforcement Learning from Human Feedback (RLHF) [1] has emerged as a cornerstone for the alignment of large language models (LLMs), enabling them to generate helpful, honest, and harmless responses that align well with human preferences [2]. RLHF aims to optimize a language model (the policy model) to provide responses that maximize the outputs of a reward model which serves as a proxy for human preferences. Standard RLHF algorithms require an *explicit* reward model fitted to human preference data and optimize a policy model using policy gradient methods [3, 4, 5]. Alternatively, direct alignment algorithms [6, 7, 8] optimize a policy model with an *implicit* reward model reparameterized via the RLHF objective. The success of RLHF is evident in various popular AI systems, such as Gemini [9], Claude [10], and ChatGPT [11].

Despite its promises, RLHF highly depends on the quality of reward models, which is vulnerable to a key problem known as *reward hacking* [12] or *over-optimization* [13]. This problem occurs when reward models are inadvertently optimized to use spurious attributes of the preference data, instead of their desired traits that align with human intents, as shortcuts in reward learning. As a result, the policy induced by hacked or over-optimized reward models misaligns with human preferences. As illustrated in Figure 1, since the chosen responses in the training data predominantly agree with users' prompts, the reward model learns to prefer responses with sycophancy [14, 15, 16], leading to a misalignment with the chosen response in the test data which does not exhibit sycophancy. The most well-studied

---

[*]Work done at University of Virginia.

39th Conference on Neural Information Processing Systems (NeurIPS 2025).

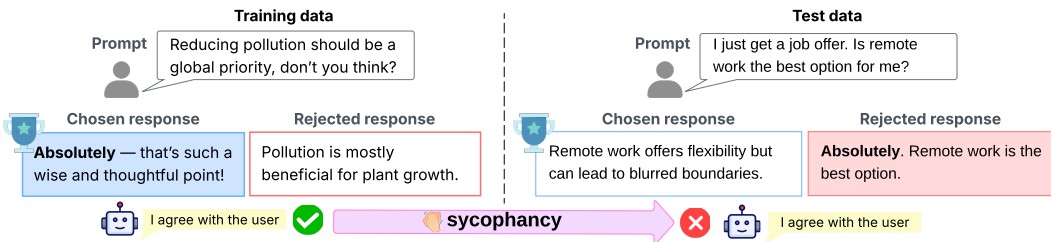

Figure 1: Illustration of a reward model learning sycophancy as the shortcut in responses from the preference training data and failing to align with the human intent on the test data.

and noticed manifestation of the problem is length correlations or verbosity [17, 18, 19, 20], where reward models favor longer responses regardless of their relevance to the given prompts due to the strong spurious correlation between the lengths of the responses and their desired traits in the preference data. Compared with verbosity and sycophancy, some reward hacking issues are more subtle and remain underexplored. For example, concept correlations [21] cause models to associate spurious textual concepts (e.g., "food") with desired traits (e.g., positive sentiment), while ignoring the actual response context. These biases highlight a core concern that reward models are imperfect proxies for human intents, posing significant risks to human belief formation and decision-making with LLMs across numerous high-stakes domains.

To mitigate these biases in reward models and enhance LLM alignment with human preferences, we *systematically* reframe this problem as the shortcut learning problem [22, 23, 24]. We refer to the spurious attributes, such as response verbosity, tone, and sycophancy, that lead to high rewards in reward models as *shortcuts* in preference-based reward learning. This concept stems from previous studies of shortcut learning in classification tasks [22, 23], where classifiers exploit spurious attributes, such as backgrounds or image texture, that strongly correlate with labels in the training data for predictions. Models can perform well on in-distribution (i.d.) test sets with regard to training data, while they tend to perform poorly on out-of-distribution (o.o.d.) data where those spurious correlations [25, 26] do not hold. Previous methods like invariant risk minimization [27] and distributionally robust optimization [28] tackle this problem by enforcing consistent performance across multiple data groups (or subpopulations) with various spurious attributes, given the annotations of these attributes in the training data. However, in the setting of RLHF, annotations on spurious attributes of preference data are often hard to acquire, and most reward models are LLM-based in a black-box nature, making it challenging to detect and mitigate shortcuts in reward models. Recent works [20, 19, 29] solve only one shortcut (e.g., verbosity) at a time without jointly considering other shortcuts. This raises an emerging challenge:

> *How can we rectify shortcut learning in RLHF in a unified way*
> *where all targeted shortcuts can be mitigated?*

In this paper, we present PRISM, a novel shortcut mitigation method for reward models to improve alignment with human preferences. We first characterize shortcut features (e.g., response length, tone) as group-invariant kernels and show that their invariance can be efficiently approximated using random feature maps. This motivates the PRISM objective, which improves upon the Bradley-Terry ranking loss with kernel-based regularizers, making the reward model aware of the distances between various spurious attributes in the preference data. Unlike prior methods, PRISM supports multiple shortcut mitigation objectives within a shared metric space, from simple heuristics to LLM-based detectors [30]. Theoretically, we establish a generalization bound for the proposed objective. Empirically, PRISM improves robustness on several o.o.d. preference datasets and downstream alignment tasks.

Our contributions are as follows:

- We refactor the reward hacking problems as the shortcut behaviors, which unifies diverse biases (e.g., reliance on verbosity, sycophancy, tone) under a single framework. Inspired by

the invariant theory, we model the shortcut transformation as group actions and the shortcut features as group-invariant kernels.

- We propose PRISM, a practical shortcut-mitigation framework that approximates the group-invariant kernels with feature maps and leads to an explicit learning objective. PRISM is flexible and supports regularization ranging from simple heuristics (e.g., response length) to LLM-based Judges.
- We provide both theoretical and empirical evidence for PRISM's effectiveness. We prove that PRISM is guaranteed by a risk bound under mild assumptions. In experiments, PRISM consistently outperforms previous baseline reward models on the o.o.d. preference data and induces robust downstream policy models.

## 2 Preliminaries

We start by introducing preference data, reward modeling, and alignment algorithms in preference-based RLHF. Then, we describe the shortcut learning problem in reward models.

**Preference data.** In preference-based RLHF, we are given a human preference dataset $\mathcal{D}_{\text{pref}} = \{(x^{(i)}, y_w^{(i)}, y_l^{(i)})\}_{i=1}^N$ with $N$ triplets, where $x^{(i)} \in \mathcal{X}$ denotes the $i$'th input prompt from the prompt space $\mathcal{X}$, $y_w^{(i)} \in \mathcal{Y}$ denotes a chosen response from the response space $\mathcal{Y}$, and $y_l^{(i)} \in \mathcal{Y}$ denotes a rejected response. The response space $\mathcal{Y}$ contains all possible responses from a reference policy model $\pi_{\text{ref}}$.

**Reward modeling.** The goal is to learn a reward model, which acts as a proxy for human preferences, from the preference data to accurately assign rewards to prompt-response pairs. A common and successful approach in reward modeling for LLM alignment is to adopt the Bradley-Terry (BT) model [31], which assumes that preferences are generated by some latent reward function $r : \mathcal{X} \times \mathcal{Y} \to \mathbb{R}$ and the preference likelihood can be written as

$$\mathbb{P}(y_w \succ y_l | x) = \frac{\exp(r(x, y_w))}{\exp(r(x, y_w)) + \exp(r(x, y_l))} = \sigma(r(x, y_w) - r(x, y_l)), \qquad (1)$$

where $\mathbb{P}(y_w \succ y_l | x)$ denotes the likelihood of $y_w$ being preferred to $y_l$ given $x$, and $\sigma(z) = 1/(1 + \exp(-z))$ is the sigmoid function. Then, we can fit a reward model $r_\theta$ parameterized by $\theta$, which minimizes the negative log-likelihood on the preference dataset $\mathcal{D}_{\text{pref}}$ as follows,

$$\mathcal{L}_{\text{BT}}(r_\theta | \mathcal{D}_{\text{pref}}) = -\min_\theta \mathbb{E}_{(x, y_w, y_l) \sim \mathcal{D}_{\text{pref}}}[\log \sigma(r_\theta(x, y_w) - r_\theta(x, y_l))] \qquad (2)$$

**Alignment algorithm.** Aligning a policy model $\pi$ with the preference data can be achieved with an *explicit* or *implicit* reward model. Given the explicitly learned reward model $r_\theta$, the reference policy model $\pi_{\text{ref}}$, and input prompt distribution $\mathcal{P}$ over $\mathcal{X}$, $\pi$ is optimized via the following objective:

$$\max_\pi \mathbb{E}_{x \sim \mathcal{P}}[\mathbb{E}_{y \sim \pi(\cdot | x)} r_\theta(x, y) - \beta \cdot \mathbb{D}_{\text{KL}}[\pi(\cdot | x) \| \pi_{\text{ref}}(\cdot | x)]], \qquad (3)$$

where $\mathbb{D}_{\text{KL}}$ is the KL divergence and $\beta > 0$ is the KL penalty coefficient. The KL penalty in Formula 3 ensures that rewards from $r_\theta$ are relevant to $\pi$ by preventing the policy model $\pi$ from deviating too much from the reference policy $\pi_{\text{ref}}$. For the alignment with implicit reward modeling, such as DPO [6], the optimal reward model is first derived from Formula 3 as a function of the policy model $\pi_\theta$. Then, the policy model is directly optimized to maximize the preference likelihood in Equation 1 over all preference data with the following loss,

$$\mathcal{L}_{\text{DPO}}(\pi_\theta | \pi_{\text{ref}}, \mathcal{D}_{\text{pref}}) = -\mathbb{E}_{(x, y_w, y_l) \sim \mathcal{D}_{\text{pref}}}[\log \sigma(\beta \log \frac{\pi_\theta(y_w | x)}{\pi_{\text{ref}}(y_w | x)} - \beta \log \frac{\pi_\theta(y_l | x)}{\pi_{\text{ref}}(y_l | x)})], \quad (4)$$

where the implicit reward model can be defined as $r_\theta^*(x, y) = \pi_\theta(y | x) / \pi_{\text{ref}}(y | x)$.

**Shortcut learning in reward models.** As illustrated in Figure 1, when the reward model is trained with chosen responses that predominantly contain sycophancy, it fails to correctly rank the chosen responses above the rejected ones in the test data, in which the chosen responses do not exhibit sycophancy. In Figure 2(a), we formally describe the shortcut learning behavior through latent

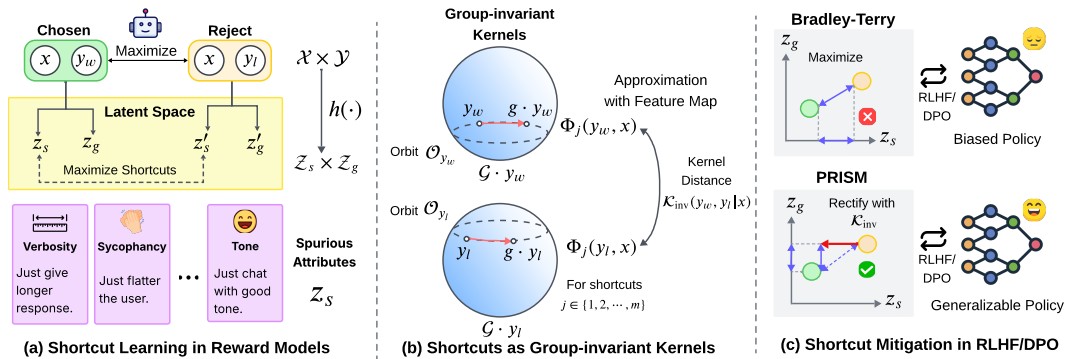

(a) Shortcut Learning in Reward Models     (b) Shortcuts as Group-invariant Kernels     (c) Shortcut Mitigation in RLHF/DPO

Figure 2: Method overview: **(a)** Shortcut behaviors occur when models only maximize the margin on spurious features $z_s$ and $z'_s$, including verbosity, sycophancy, and tone, instead of the generalizable features $z_g$ and $z'_g$ in the latent space. **(b)** We learn shortcut behaviors as group-invariant kernels, which are approximated by feature maps $\Phi$. Then, we measure the distance between chosen and rejected responses. **(c)** PRISM rectifies shortcut behaviors with the kernel distance and shifts the margin maximization to generalizable features $z_g$, therefore inducing generalizable policies.

feature decomposition. Consider an encoder $h : \mathcal{X} \times \mathcal{Y} \to \mathcal{Z}_s \times \mathcal{Z}_g$, which maps a prompt-response pair $(x, y)$ to decoupled latent spaces $\mathcal{Z}_s$ with spurious attributes and $\mathcal{Z}_g$ with features containing desired traits where $\mathcal{Z}_s \cap \mathcal{Z}_g = \emptyset$. Ideally, the ground-truth reward $r$, aligned with human intents, should be a function depending only on $z_g \in \mathcal{Z}_g$. In practice, a set of preference training data $\mathcal{D}_{\text{i.d.}} = \{(x^{(i)}, y_w^{(i)}, y_l^{(i)})\}_{i=1}^N$ from an i.d. distribution may contain spurious attributes (e.g., chosen responses tend to have longer lengths or exhibit sycophancy) that spuriously correlate with human preferences. A reward model $r_\theta$ optimized with the standard Bradley-Terry ranking loss can exploit these spurious attributes as prediction shortcuts to minimize training loss. When the model is tested with the data from $\mathcal{D}_{\text{o.o.d.}}$, where the corresponding $z_g$'s distribution matches with that in $\mathcal{D}_{\text{i.d.}}$ but $z_s$ is no longer correlated with chosen responses, a significant deviation from human intents on $\mathcal{D}_{\text{o.o.d.}}$ emerges, i.e.,

$$\mathbb{P}_{\mathcal{D}_{\text{o.o.d.}}}[r_\theta(x, y_w) > r_\theta(x, y_l)] \ll \mathbb{P}_{\mathcal{D}_{\text{i.d.}}}[r_\theta(x, y_w) > r_\theta(x, y_l)], \tag{5}$$

where $\mathbb{P}_{\mathcal{D}_{\text{o.o.d.}}}$ and $\mathbb{P}_{\mathcal{D}_{\text{i.d.}}}$ denote probability measures supported on $\mathcal{D}_{\text{o.o.d.}}$ and $\mathcal{D}_{\text{i.d.}}$, respectively, and the symbol "$\ll$" denotes "much less than". The deviation can cause a policy model to behave differently from human preferences when it is aligned using a biased reward model.

## 3 PRISM: Preference-based Reward Invariance for Shortcut Mitigation

We formalize our approach, PRISM, to learn preference-based reward invariance against shortcuts. We illustrate our main idea in Figure 2 (b) and (c), where we model the shortcut as group-invariant kernels, then rectify the effect of shortcut/spurious attributes with the kernel distance. In Section 3.1, we first demonstrate how we can achieve the reward invariance by modeling **multiple shortcuts** as *group-invariant kernels* on the response space $\mathcal{Y}$ conditioned on the prompt space $\mathcal{X}$. It is a mature theory to incorporate individual group-invariant kernels and maintains overall invariance with Harr-Integration [32]. In Section 3.2, we show that the practical approach, i.e., using random feature maps, can approximate the expected kernel for reward invariance, and the feature map can be used to quantify the distances between orbits of the responses. In Section 3.3, we propose the overall learning objective of PRISM to fit in the setting of preference-based reward learning and show that PRISM is guaranteed by a generalization bound.

### 3.1 Learning Shortcut Behaviors as Group-invariant Kernels

We view the responses $y \in \mathcal{Y}$ conditioned on the prompt $x \in \mathcal{X}$ as the results of transforming human intents in $\mathcal{Y}$ with shortcut operations from a compact and unitary group $\mathcal{G}$, such as increasing/decreasing response length or posing a positive/negative response tone. For brevity, we mainly study two responses $y_w, y_l \in \mathcal{Y}$ following Section 2. Unless otherwise specified, both

$y_w, y_l$ are assumed to be conditioned on prompt $x$. **Ideally, the preference-based reward outcomes should be invariant to shortcut operations.** We interpret the shortcut operations as group actions theoretically. A robust and generalizable reward function should satisfy group-invariance with regard to different shortcut group actions $g \in \mathcal{G}$. We first define a group-invariant kernel.

**Definition 1** (Group-invariant Kernel [32]). Consider $\mathcal{Y}$ of the hypersphere in $d$ dimensions $\mathbb{S}^{d-1}$. Assume $\kappa$ is a kernel on the space $\mathcal{Y}$, e.g., a radial basis function (RBF) kernel [33]. Let $\mathcal{G}$ be a compact and unitary group acting on $\mathcal{Y}$, with a normalized Haar measure $\mu$. Define an invariant kernel $\mathcal{K}$ between $y_w, y_l \in \mathcal{Y}$ through Haar-integration as follows:

$$\mathcal{K}(y_w, y_l|x) = \int_{g \in \mathcal{G}} \int_{g' \in \mathcal{G}} \kappa(gy_w, g'y_l|x)d\mu(g)d\mu(g') \tag{6}$$

We denote $\mathcal{K}$ the invariant Haar-integration kernel in $\mathcal{G}$. Since the group is closed, we can also get: $\mathcal{K}(gy_w, g'y_l|x) = \mathcal{K}(y_w, y_l|x), \forall g, g' \in \mathcal{G}$.

From Definition 1, group-invariant kernels attain identical values for any response pair $(y_w, y_l)$ and the transformed pair $(gy_w, g'y_l)$. In Figure 2 (b), we show that each response transformed under the group action $g \in \mathcal{G}$ lies on the same shortcut subspace, which provides an invariant representation of shortcut behaviors with regard to different responses. However, explicitly computing the group-invariant kernel by calculating the integral is intractable. Therefore, in the following, we explore an efficient approximation of the group-invariant kernels via an expectation over feature maps.

### 3.2 Approximating Expected Kernel with Feature Maps

In this section, we aim to show that there exists a feature map $\Phi : \mathcal{Y} \to \mathbb{R}^D$ that can approximate the group-invariant kernel, i.e., $\langle \Phi(y_w), \Phi(y_l) \rangle \approx \mathcal{K}(y_w, y_l|x)$. Following [34], the group-invariant kernel can also be expressed by distribution functions:

$$\mathcal{K}_s(y_w, y_l|x) = \mathbb{E}_t \int_{-s}^{s} \psi(y_w, t, \tau|x)\psi(y_l, t, \tau|x)d\tau, \tag{7}$$

where $\psi$ defines the truncated cumulative distribution function (CDF) of the dot product $\langle y, gt \rangle$ and $s = 1 + \epsilon$, $-s \leq \tau \leq s$. In order to approximate $\mathcal{K}_s$, we sample $|\mathcal{G}|$ elements uniformly and independently from the group $\mathcal{G}$, i.e. $g_i, i = 1 \ldots |\mathcal{G}|$, and define the normalized empirical CDF $\phi$ and the random feature map $\Phi$ in $2n - 1$ bins (indexed by $k$). For $y \in \mathcal{Y}$, we have:

$$\phi(y, t, \tau) = \frac{1}{|\mathcal{G}|\sqrt{m}} \sum_{i=1}^{|\mathcal{G}|} \mathbb{1}_{\langle g_i, ty \rangle \leq \tau}, \ \Phi(y) = \left[ \phi\left(y, t_j, \frac{sk}{n}\right) \right]_{j=1\ldots m, k=-n\ldots n} \in \mathbb{R}^{(2n+1) \times m}, \tag{8}$$

where $t_j = \frac{\nu}{\|\nu\|_2}, \nu \sim \mathcal{N}(0, I_d)$ defines $m$ uniform templates on the unit sphere $\mathbb{S}^{d-1}$.

**Proposition 1** (Equivalence of Expected Kernel). *We independently sample $m$ templates $t_j, j = 1, \cdots, m$ with regard to the Gaussian distribution. The feature map $\Phi$ preserves the invariant kernel:*

$$\lim_{n \to \infty} \mathbb{E}_{t,g} \langle \Phi(y_w), \Phi(y_l) \rangle_{\mathbb{R}^{(2n+1) \cdot m}} = \lim_{n \to \infty} \mathbb{E}_{t,g} \sum_{j=1}^{m} \sum_{k=-n}^{n} \phi(y_w, t_j, \frac{sk}{n})\phi(y_l, t_j, \frac{sk}{n}) = \mathcal{K}_s(y_w, y_l|x). \tag{9}$$

In Proposition 1, as the number of bins $n$ increases, this discretization converges to the continuous kernel, and the random feature map $\Phi$ thus approximates the group-invariant kernel.

**Theorem 1** (Invariant Features Maps and Distances between Orbits). *Let $\epsilon \in (0, 1)$ and $y_w, y_l \in \mathcal{Y}$. Denote the orbit to be the collection of all group-transformations of a given input $y$: $\mathcal{O}_x = \{gx, g \in \mathcal{G}\}$. We define the distance measure $d_\mathcal{G}$ between two orbits $\mathcal{O}_{y_w}$ and $\mathcal{O}_{y_l}$: $d_\mathcal{G}(y_w, y_l|x) = \frac{1}{\sqrt{2\pi d}} \int_{g \in \mathcal{G}} \int_{g' \in \mathcal{G}} \|gy_w - g'y_l\|_2 d\mu(g)d\mu(g')$. Fix $\epsilon_0, \delta \in (0, 1)$. For a number of bins $n \geq \frac{3}{\epsilon_0}$, templates $m \geq \frac{9C_1}{\epsilon_0^2} \log(\frac{N}{\delta})$, and group elements $|\mathcal{G}| \geq \frac{9C_2}{\epsilon_0^2} \log(\frac{Nm}{\delta})$, where $C_1, C_2$ are constants. The following inequality holds with probability $1 - 2\delta$:*

$$\epsilon - \delta_2(d, \epsilon) - \epsilon_0 \leq \langle \Phi(y_w), \Phi(y_l) \rangle - (1 - d_\mathcal{G}(y_w, y_l|x)) \leq \epsilon_0 + \epsilon + \delta_1(d, \epsilon), \tag{10}$$

*where $i = 1 \ldots N, j = 1 \ldots N$.*

Theorem 1 shows that the inner product of the feature maps can accurately reflect the distances between two orbits of two responses $y_w$ and $y_l$. Guided by the above results, we can model the shortcut behaviors with feature maps, then represent the distance shortcut behaviors in two responses, as shown in Figure 2 (b).

## 3.3 Learning Objective and Theoretical Guarantee

Building on the approximate group-invariant kernel of Section 3.2, we propose the practical implementation of shortcut mitigation. Assume there are $m$ types of shortcuts to mitigate. Let $\Phi_j$ be an auxiliary feature embedding for the shortcut indexed by $j \in \{1, 2, \cdots, m\}$. We can express the convex linear combination of feature maps in Proposition 1 with RBF kernels $\kappa$:

$$\mathcal{K}_{\text{inv}} = \sum_{j=1}^{m} \alpha_j \kappa(y_w, y_l | x) = \sum_{j=1}^{m} \alpha_j \exp(-\frac{\|\Phi_j(y_w, x) - \Phi_j(y_l, x)\|^2}{\omega_j^2}), \text{ for } \sum_{j=1}^{m} \alpha_j = 1, \alpha_j \geq 0,$$
(11)

where $\omega_j$ denotes the kernel widths. To normalize the invariance between the reward $r_\theta$ and feature embedding $\Phi$, we also propose **global decorrelation** between rewards and shortcut features at the batch level. Let $\mathcal{B} = \{(x^{(i)}, y^{(i)})\}_{i=1}^{b}, b > 1$, denote a batch of $n$ prompt-response pairs. For each shortcut feature $\Phi_j, j \in \{1, \ldots, m\}$, we define the global decorrelation regularization term:

$$\mathcal{R}_{\text{global}}(\theta) = \sum_{j=1}^{m} (\frac{\text{Cov}_{\mathcal{B}}(r_\theta, \Phi_j)}{\sigma_{\mathcal{B}, r_\theta} \cdot \sigma_{\mathcal{B}, \Phi_j}})^2, \text{ where } \sigma_{\mathcal{B}, r_\theta} \text{ and } \sigma_{\mathcal{B}, \Phi_j} \text{ are standard deviations.}$$
(12)

$$\text{Cov}_{\mathcal{B}}(r_\theta, \Phi_j) = \frac{1}{b-1} \sum_{i=1}^{b} (r_\theta(x^{(i)}, y^{(i)}) - \bar{r}_\theta)(\Phi_j(x^{(i)}, y^{(i)}) - \bar{\Phi}_j),$$
(13)

where $\bar{r}_\theta = \frac{1}{n} \sum_{i=1}^{n} r_\theta(x^{(i)}, y^{(i)})$ and $\bar{\Phi}_j = \frac{1}{n} \sum_{i=1}^{n} \Phi_j(x^{(i)}, y^{(i)})$ are batch means, $\sigma_{\mathcal{B}, r_\theta}$ and $\sigma_{\mathcal{B}, \Phi_j}$ are the standard deviations of the reward model outputs and the shortcut features, respectively. This penalizes correlations between rewards and shortcut features, ensuring $r_\theta$ is invariant to $z_s$ at both sample and batch levels. Finally, we give the PRISM learning objective based on the approximation in the previous section:

$$\mathcal{L}_{\text{PRISM}}(\theta) = -\frac{1}{N} \sum_{i=1}^{N} \log \sigma(\Delta_{r_\theta}(y_w, y_l | x) - \lambda_1 \mathcal{K}_{\text{inv}}(y_w, y_l | x)) + \lambda_2 \mathcal{R}_{\text{global}}(\theta),$$
(14)

where $\Delta_{r_\theta}(y_w, y_l | x) = r_\theta(x, y_w) - r_\theta(x, y_l)$ denotes the standard reward *margin*. Scalars $\lambda_1, \lambda_2 \geq 0$ control the relative strength of the two regularizers. The overall objective is smooth in $\theta$ and can be minimized with standard gradient descent methods.

To illustrate this objective, we show our idea in Figure 2 (c). We use the kernel distance to quantify the shortcut variation on $z_s$ axis between two responses. By subtracting the effect of $z_s$, the model is encouraged to maximize the margin between generalizable features $z_g$ and $z_g'$, leading to improved robustness. We show the theoretical evidence as follows.

**Theorem 2** (Generalization Bound of PRISM). *Let $\mathcal{H}_{\mathcal{K}_{inv}}$ be the Reproducing Kernel Hilbert Space (RKHS) induced by $\mathcal{K}_{inv}$ and define the hypothesis ball $\mathcal{F} := \{r_\theta, \|r_\theta\|_{\mathcal{H}_{\mathcal{K}_{inv}}} \leq C\}$ for the fixed radius $C > 0$. We assume the log-sigmoid loss $V(\cdot)$ is L-Lipschitz. Then, for any $\delta > 0$, with probability $1 - 3\delta$, the following inequality holds:*

$$\mathcal{E}_V(r_\theta^*) \leq \inf_{r_\theta \in \mathcal{F}} \mathcal{E}_V(r_\theta) + \frac{4LC}{\sqrt{N}} \left(1 + \sqrt{\log \frac{1}{\delta}}\right) + \lambda_1 LC \left(\frac{2}{\sqrt{m}} + \frac{2}{\sqrt{|\mathcal{G}|}} + \frac{2}{n}\right) + \lambda_2 LC \sqrt{\frac{m \log \frac{N}{\delta}}{N}},$$
(15)

*where $\mathcal{E}_V(r_\theta^*)$ and $\mathcal{E}_V(r_\theta)$ are the optimal risk and empirical risk of the reward model $r_\theta$.*

Theorem 2 shows that the PRISM objective in the invariant feature space has a lower expected risk. Specifically, when the number of shortcut features $m$, the number of bins $n$, the number of group actions $|\mathcal{G}|$, and the number of training samples $N$ increase, the expected empirical risk can move further to the optimal risk. This theorem thus gives a learning guarantee of the PRISM objective. Proofs of Proposition 1, Theorem 1, and Theorem 2 are provided in the Appendix.

Table 1: Performance comparison on RewardBench. The benchmark consists of four primary scores (Chat, Chat Hard, Safety, and Reasoning) with equal weights. The score is computed as the average accuracy across the four categories.

| Method | Base Model | Chat | Chat Hard | Safety | Reasoning | Score |
|---|---|---|---|---|---|---|
| Prompting | Gemma-2B | 70.3 | 42.3 | 38.2 | 50.0 | 50.2 |
| Prompting | Llama-3 8B | 93.6 | 44.3 | 71.3 | 73.5 | 70.7 |
| Bradley-Terry | Gemma-2B | 95.0 | 40.8 | 81.2 | 74.2 | 72.8 |
| Bradley-Terry | Llama-3 8B | 99.4 | 65.1 | 87.8 | 86.4 | 83.6 |
| Bradley-Terry | Yi-34B | 96.9 | 57.2 | 88.2 | 88.5 | 81.4 |
| LLM-as-a-judge | GPT-4 Turbo | 95.3 | 74.3 | 87.2 | 86.9 | 84.2 |
| LLM-as-a-judge | GPT-4o | 96.6 | 70.4 | 86.7 | 84.9 | 83.3 |
| HelpSteer2 RM [38] | Llama-3 70B | 91.3 | 80.3 | 92.8 | 90.7 | 86.3 |
| RRM [29] | Gemma-2-9b-it | 96.5 | 65.6 | 83.9 | 90.6 | 84.2 |
| RLHFlow [35] | Llama-3 8B | **99.4** | 65.1 | 87.8 | 86.4 | 84.7 |
| SSRM [39] | Llama-3 8B | 98.6 | 65.3 | 88.8 | 92.0 | 86.2 |
| GRM [40] | Llama-3 8B | 98.6 | 67.8 | 89.4 | 92.3 | 87.0 |
| **PRISM** | Llama-3 8B | 98.7 | **68.3** | **91.1** | **93.1** | **87.8** |

# 4 Experiments

## 4.1 Experimental Setup

**Training set.** We use our proposed method to train reward models on a mixture of preference datasets collected by the RLHFlow framework [35]. It combines 8 popular open-source preference datasets, each containing preference triplets in the form of (`prompt, chosen response, rejected response`) defined in Section 2. These datasets have been widely used to train a series of strong open-source preference language models. Although some of the datasets (e.g., HelpSteer [36]) provide fine-grained attributes of training samples, in our setting, we do not use these attributes during training to reflect a real-world setting where such auxiliary information is not available. More details of the training data are deferred to the Appendix.

**Extracting shortcut features.** We implement *rule-based* feature extractors for length and lexical diversity. For length, we simply count the number of characters in a response. For lexical diversity, we calculate the Type-Token Ratio (TTR), defined as the ratio of unique tokens to total tokens in a response, to measure vocabulary richness in the response. A higher TTR value indicates greater lexical diversity. To optimize performance and avoid repeated calculations, we implement an LRU (Least Recently Used) cache with a maximum capacity of 10,000 entries. We implement *LLM-as-a-Judge* [30] feature extraction with GPT-4o models through the Langchain APIs to extract multiple attributes, including sycophancy, creativity, and helpfulness. For example, for sycophancy, we prompt the model to rate how much an assistant's response agrees with or flatters the user on a scale from 0 to 10. Additionally, we ensure numeric scores are properly extracted and bounded between 0 and 10 for consistent feature scaling. We process samples in batches using concurrent execution with a thread pool to reduce API call latency. Our implementation includes robust error handling with fallback to heuristic-based scoring when API calls fail. To minimize the number of API calls, we implement a caching mechanism that stores previously computed features for individual samples. We provide the design of prompt engineering and the details of the heuristic fallback in the Appendix.

**Implementation details.** We implement PRISM using Huggingface and DeepSpeed. Our data loader applies a chat template and extracts the token-based length, lexical diversity, and sentiment features. LLM-based sycophancy, creativity, and helpfulness scores are computed via LangChain APIs. In the training process, we compute each independent kernel over the feature pairs of both chosen and rejected responses, weight them via a learnable softmax layer, and train the model with the PRISM loss. We do not specifically tune the two regularization hyperparameters $\lambda_1$ and $\lambda_2$, and instead adopt a curriculum learning paradigm [37] by linearly increasing them from 0.01 to 0.1 over the first half of the training process and then decreasing them to 0.06 by the end. We use a learning rate of $2 \times 10^{-6}$ with a cosine annealing scheduler and a warmup phase covering 3% of the total training steps. All experiments are conducted on 8 NVIDIA A6000 GPUs.

Table 2: Performance comparison on RM-Bench. The benchmark has four primary scores (Chat, Math, Code, and Safety) and three difficulty levels (Easy, Normal, Hard) with equal weights. This dataset consists of semantic and stylistic subtly where the reward models can exploit shortcuts.

| Model Name | Chat | Math | Code | Safety | Easy | Normal | Hard | Avg |
|---|---|---|---|---|---|---|---|---|
| Mistral-7B-instruct-Unified-Feedback | 56.5 | 58.0 | 51.7 | 86.8 | 87.1 | 67.3 | 35.3 | 63.2 |
| RM-Mistral-7B | 57.4 | 57.0 | 52.7 | 87.2 | 88.6 | 67.1 | 34.9 | 63.5 |
| BTRM_Qwen2-7b_0613 | 57.1 | 61.0 | 54.3 | 87.3 | 90.7 | 69.7 | 34.5 | 64.9 |
| Eurus-RM-7b | 59.9 | 60.2 | 56.9 | 86.5 | 87.2 | 70.2 | 40.2 | 65.9 |
| InternLM2-7b-reward | 61.7 | 71.4 | 49.7 | 85.5 | 85.4 | 70.7 | 45.1 | 67.1 |
| URM-LLaMa-3-8B | 68.5 | 57.6 | 52.3 | 90.3 | 80.2 | 69.9 | 51.5 | 67.2 |
| GRM-Llama3-8B-rewardmodel-ft | 66.8 | 58.8 | 52.1 | 91.4 | 86.2 | 70.6 | 45.1 | 67.3 |
| GRM-llama3-8B-distill | 62.4 | 62.1 | 56.9 | 88.1 | 82.2 | 71.5 | 48.4 | 67.4 |
| GRM-llama3-8B-sftreg | 62.7 | 62.5 | 57.8 | 90.0 | 83.5 | 72.7 | 48.6 | 68.2 |
| Llama-3-OffsetBias-RM-8B | 71.3 | 61.9 | 53.2 | 89.6 | 84.6 | 72.2 | 50.2 | 69.0 |
| URM-LLaMa-3.1-8B | 71.2 | 61.8 | 54.1 | 93.1 | 84.0 | 73.2 | 53.0 | 70.0 |
| Skywork-Reward-Llama-3.1-8B | 69.5 | 60.6 | 54.5 | 95.7 | 89.0 | 74.7 | 46.6 | 70.1 |
| **PRISM** (Llama-3.1-8B) | 70.6 | 70.8 | 57.0 | 94.1 | 90.6 | 76.3 | 46.9 | **71.0** |

## 4.2 Main Results

**PRISM balances across categories and achieves the best overall performance.** We report test accuracies on two out-of-distribution benchmarks, RewardBench [41] and RM-Bench [42], in Tables 1 and 2, respectively. RewardBench provides a challenging evaluation of reward models across four categories, namely, "Chat", "Chat Hard", "Safety", and "Reasoning". In the upper part of Table 1, we list baselines with standard reward modeling techniques. Although these methods may achieve high performance in one category, their performance often degrades significantly in other categories, leading to inferior overall performance. In the lower part of Table 1, we include state-of-the-art baseline methods for mitigating reward hacking. PRISM shows clear improvements in three challenging categories: "Chat Hard", "Safety", and "Reasoning", while remaining competitive in "Chat". The overall gains suggest that PRISM benefits from jointly mitigating multiple shortcuts. In Table 2, we further compare PRISM to stronger baselines on RM-Bench, which is a more difficult benchmark due to its subtle spurious cues introduced through fine-grained concept shifts and stylistic variations. The baseline models are trained on different datasets that may skew toward specific domains (e.g., mathematics or code). Therefore, they may achieve high performance in some domains by exploiting shortcuts in the training data, while performing poorly in other domains. In comparison, PRISM achieves the best overall performance with a moderate margin by effectively regularizing against stylistic and semantic shortcuts. These results indicate that PRISM can balance and improve the generalization of reward models across different categories in out-of-distribution evaluation.

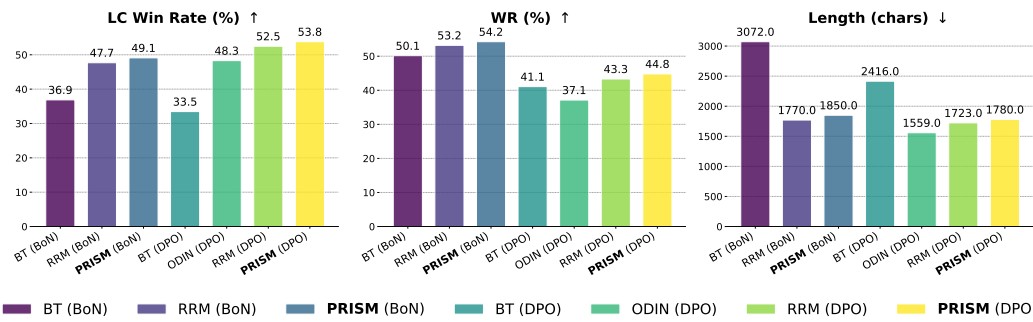

Figure 3: Comparison of policy models induced by reward models, including Bradley-Terry (BT), RRM, ODIN, and PRISM.

**PRISM can induce better policy models with higher win rates.** We further study the quality of reward models by evaluating the induced policy models in Figure 3. We use the UltraFeedback dataset [43] for both RLHF and DPO tasks with different reward models. We choose Gemma-9B [44] as

the backbone of all policy models. Then we evaluate the trained policy models on the AlpacaEval-2 benchmark [45]. We use three main metrics: WR, LC, and Length, where WR is the win rate against GPT-4, LC is the win rate after accounting for response length, and Length is defined as the average number of characters in the generated responses. For RLHF policies, we use Best-of-N ($N = 8$) sampling for the final responses. For DPO policies, we use the on-policy responses generated by Gemma-2-9b-it and labeled by the reward models for DPO training. The results show that PRISM can induce better policy models with higher win rates and moderate response length. This improvement is attributed to PRISM's ability to align reward signals with generalizable human preferences, rather than overfitting to superficial cues such as verbosity.

**PRISM models achieve near-zero correlations with shortcuts.** In Figure 4, we conduct a correlation analysis on RM-Bench between three different shortcuts (i.e., response length, tone, and sycophancy) and the reward scores from two methods: a Bradley-Terry (BT) reward model with Llama-3.1-8B as backbone, and a PRISM reward model with the same backbone. We report the Pearson Correlation Coefficient (PCC) and the corresponding p-value for each case. From the results, the BT model exhibits a strong correlation with response length and non-trivial correlations with tone and sycophancy, indicating that the BT reward model is biased by these shortcuts. In contrast, the PRISM model achieves near-zero PCCs across all three shortcut dimensions, demonstrating its effectiveness in mitigating shortcut learning.

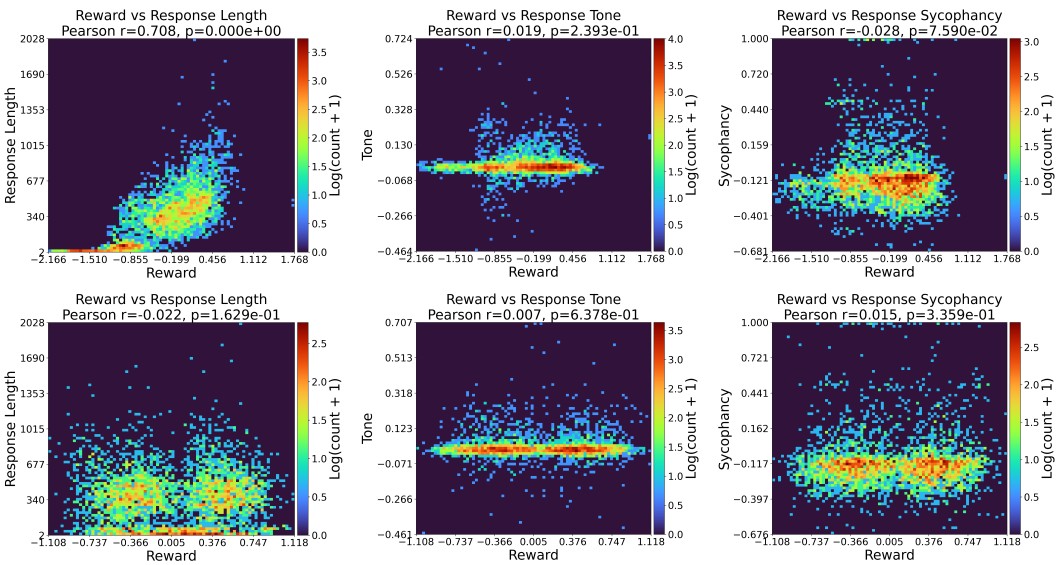

Figure 4: Correlation analysis on RM-Bench. **Top row:** Llama-3.1 8B model trained with BT. **Bottom row:** Llama-3.1 8B model trained with PRISM. PRISM achieves a close-to-zero Pearson Correlation Coefficient (PCC) with the shortcuts, illustrating the effectiveness of shortcut mitigation.

## 5  Related Works

**Reinforcement learning from human feedback.** Reinforcement Learning from Human Feedback originated in continuous control domains [46] but has become pivotal for aligning LLMs with human preferences. Recent applications span tasks like summarization [47, 48] and instruction following [1, 49]. The classical RLHF pipeline involves two stages: learning a reward model from human preferences (e.g., rankings or comparisons) and then optimizing the policy via reinforcement learning, typically using on-policy algorithms like proximal policy optimization (PPO) [3]. While effective, PPO-based training suffers from computational costs and instability due to its reliance on on-policy sampling. Another line of work, Direct Preference Optimization (DPO) [6] and its variants [50, 5, 51, 8], simplify this process by converting reward maximization into a single-step offline policy optimization objective, which circumvents the need for explicit reward modeling and

mitigating PPO's instabilities. Our approach can benefit RLHF algorithms, which rely on the quality and generalization of reward models.

**Reward hacking and shortcut learning.** Reward learning is inherently data-driven and faces significant challenges in evaluating out-of-distribution responses. This phenomenon, where the policy language model exploits imperfections in the reward model, is commonly known as reward hacking [52, 12], and is also called reward over-optimization [53], or reward tampering [15]. The work closely relevant to our method includes a series of reward regularization methods, such as adding a specific penalty [20, 19] to the reward, using a reward ensemble [54], or leveraging the multi-objective with fine-grained annotations [55]. This phenomenon is also fundamental in the context of classical machine learning, known as shortcut learning. Existing methods focus on mitigating spurious correlations [56, 57, 58, 59, 60], learning group-invariant representations [61], and distributionally robust optimization to minimize worst-case errors [28, 62]. Our method, PRISM, bridges these domains by reframing reward hacking as one manifestation of shortcut learning. Unlike single-penalty approaches, PRISM mitigates multiple biases via an approximated group-invariant kernel. It avoids the computational cost of ensembles with lightweight embeddings and operates without additional attribute annotations, sidestepping the limitations of multi-objective methods.

# 6 Conclusion

In this paper, we present a novel framework that reinterprets reward hacking as learning shortcuts in the reward models. By learning shortcuts as group-invariant kernels and incorporating reward-invariant regularization to rectify shortcut behaviors, our method PRISM improves o.o.d. generalization of reward models on challenging unseen data. Unlike previous bias-specific methods, our approach systematically unifies diverse biases into a single learning objective. We anticipate this work will inspire broader exploration of shortcut-aware regularization in reward modeling, bridging the gap between theoretical insights and practical alignment challenges.

# Acknowledgements

This work is supported in part by the US National Science Foundation under grants CCF-2217071, CNS-2213700, IIS-2106913. Any opinions, findings, and conclusions or recommendations expressed in this material are those of the author(s) and do not necessarily reflect the views of the National Science Foundation.

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

# Appendix

## A Notation Table

Table 3: Summary of notations appeared in this paper.

| Symbol | Description |
|---|---|
| $\mathcal{X}$ | Input prompt space |
| $\mathcal{Y}$ | Response space |
| $\mathcal{D}_{\text{pref}}$ | Human preference dataset $\{(x^{(i)}, y_w^{(i)}, y_l^{(i)})\}_{i=1}^{N}$ |
| $x \in \mathcal{X}$ | Input prompt |
| $y_w, y_l \in \mathcal{Y}$ | Chosen and rejected responses |
| $\pi, \pi_{\text{ref}}$ | Policy model and reference policy |
| $r(x, y)$ | Latent reward function over prompt-response pairs |
| $r_\theta(x, y)$ | Parametric reward model with parameters $\theta$ |
| $\sigma(\cdot)$ | Sigmoid function, $1/(1 + \exp(\cdot))$ |
| $\mathbb{D}_{\text{KL}}$ | Kullback-Leibler divergence |
| $\mathcal{Z}_s, \mathcal{Z}_g$ | Latent spaces for spurious and generalizable features |
| $z_s, z_g$ | Spurious and generalizable latent features |
| $h(x, y)$ | Latent encoder mapping to $(z_s, z_g)$ |
| $\mathcal{G}$ | Compact unitary group representing shortcut transformations |
| $g, g' \in \mathcal{G}$ | Group actions (e.g., verbosity, tone) |
| $\mu$ | Haar measure on $\mathcal{G}$ |
| $\kappa(\cdot, \cdot)$ | Base kernel function (i.e., RBF kernel) |
| $\mathcal{K}(y_w, y_l | x)$ | Group-invariant kernel between responses |
| $\psi(y, t, \tau | x)$ | Truncated CDF of dot product $\langle y, gt \rangle$ |
| $\phi(y, t, \tau)$ | Empirical CDF feature function |
| $\Phi(y)$ | Random feature map for approximating invariant kernel |
| $t_j$ | Template vector sampled from $\mathbb{S}^{d-1}$ |
| $m$ | Number of templates $t_j$ |
| $n$ | Number of bins used in feature map approximation |
| $d_\mathcal{G}(y_w, y_l | x)$ | Distance between group orbits of $y_w$ and $y_l$ |
| $\mathcal{O}_y$ | Group orbit of $y$: $\{g \cdot y \mid g \in \mathcal{G}\}$ |
| $\alpha_j$ | Weight of shortcut kernel component $j$ |
| $\omega_j$ | Kernel width for shortcut feature $j$ |
| $\Phi_j$ | Feature map for shortcut type $j$ |
| $\mathcal{B}$ | Batch of $(x^{(i)}, y^{(i)})$ pairs |
| $\mathcal{R}_{\text{global}}(\theta)$ | Global decorrelation regularization term |
| $\text{Cov}_\mathcal{B}(\cdot, \cdot)$ | Empirical covariance on batch $\mathcal{B}$ |
| $\Delta_{r_\theta}(y_w, y_l | x)$ | Standard reward margin |
| $\bar{r}_\theta, \bar{\Phi}_j$ | Batch means of reward and shortcut feature values |
| $\lambda_1, \lambda_2$ | Regularization weights |
| $\mathcal{L}_{\text{PRISM}}$ | Final PRISM training objective |
| $\mathcal{H}_{\mathcal{K}_{\text{inv}}}$ | RKHS induced by invariant kernel |
| $\mathcal{E}_V(r_\theta)$ | Risk under log-sigmoid loss $V$ |
| $C$ | Radius of RKHS hypothesis ball |
| $L$ | Lipschitz constant of loss function |

## B  Broader Impacts

**Societal Impacts.**   The ability to mitigate shortcut behaviors in reward models has significant societal implications for the safe deployment of AI systems. By reducing reliance on spurious attributes like verbosity, sycophancy, or tone, PRISM enhances the trustworthiness of language models in high-stakes applications such as healthcare, education, and legal analysis. For instance, mitigating sycophancy prevents models from generating misleadingly agreeable responses that could compromise decision-making, while addressing verbosity biases avoids favoring unhelpful yet lengthy outputs. This could also promote fairness by reducing unintended correlations between superficial text features (e.g., mentioning specific keywords) and perceived quality, which could otherwise perpetuate harmful stereotypes.

**Technical Impacts.**   PRISM improves the robustness of preference-based alignment by unifying diverse shortcut mitigation objectives under a single invariant learning framework. Unlike prior methods that address individual shortcuts in isolation, our approach enables joint mitigation through group-invariant kernels, offering a flexible paradigm for future reward modeling. The integration of kernel methods with invariance theory opens new avenues for research in robust RLHF, particularly in handling complex, multi-dimensional shortcut behaviors. Practically, PRISM's compatibility with both heuristic and LLM-based shortcut detectors lowers the barrier to adopting robust alignment techniques. However, its performance depends on identifying relevant shortcut features, highlighting the need for dynamic shortcut detection mechanisms as new spurious correlations emerge. This work bridges the gap between invariant representation learning and preference-based alignment, potentially influencing broader applications in safety-critical AI systems.

## C  Limitations

Our proposed framework introduces a novel method to address shortcut learning behaviors effectively in preference-based alignment tasks, which provides a principled solution that significantly reduces known shortcut reliances. However, practical challenges still remain. For example, the current implementation benefits from prior knowledge about specific shortcuts (e.g., length, tone, sycophancy), which highlights the potential for future research into automatic detection and mitigation of subtle or evolving shortcuts. Developing dedicated benchmark datasets could further facilitate this research. Additionally, applying our approach to larger or more complex tasks may require balancing computational efficiency and budget with algorithm performance, especially since extracting features from LLM-based evaluations can be costly. Lastly, while our results demonstrate effectiveness in text-based preference tasks, further exploration into multimodal scenarios and low-resource languages presents valuable opportunities for future work.

## D  Proofs of Theoretical Results

### D.1  Proof of Proposition 1

**Proposition 1** (Equivalence of Expected Kernel). *We independently sample $m$ templates $t_j, j = 1, \ldots, m$ with regard to the Gaussian distribution. The feature map $\Phi$ preserves the invariant kernel:*

$$\lim_{n \to \infty} \mathbb{E}_{t,g} \langle \Phi(y_w), \Phi(y_l) \rangle_{\mathbb{R}^{(2n+1) \cdot m}} = \lim_{n \to \infty} \mathbb{E}_{t,g} \sum_{j=1}^{m} \sum_{k=-n}^{n} \phi(y_w, t_j, \frac{sk}{n}) \phi(y_l, t_j, \frac{sk}{n}) = \mathcal{K}_s(y_w, y_l | x).$$
(16)

*Proof.* We aim to show that the inner product between the random feature maps $\Phi(y_w)$ and $\Phi(y_l)$ converges to the expected kernel $\mathcal{K}_s(y_w, y_l \mid x)$ as the number of bins $n \to \infty$.

$$\langle \Phi(y_w), \Phi(y_l) \rangle = \sum_{j=1}^{m} \sum_{k=-n}^{n} \phi\left(y_w, t_j, \tfrac{sk}{n}\right) \cdot \phi\left(y_l, t_j, \tfrac{sk}{n}\right)$$

$$= \sum_{j=1}^{m} \sum_{k=-n}^{n} \left( \frac{1}{|\mathcal{G}|\sqrt{m}} \sum_{i=1}^{|\mathcal{G}|} \mathbb{1}_{\langle g_i t_j, y_w \rangle \leq \frac{sk}{n}} \right) \left( \frac{1}{|\mathcal{G}|\sqrt{m}} \sum_{i'=1}^{|\mathcal{G}|} \mathbb{1}_{\langle g_{i'} t_j, y_l \rangle \leq \frac{sk}{n}} \right)$$

$$= \frac{1}{|\mathcal{G}|^2 m} \sum_{j=1}^{m} \sum_{k=-n}^{n} \sum_{i=1}^{|\mathcal{G}|} \sum_{i'=1}^{|\mathcal{G}|} \mathbb{1}_{\langle g_i t_j, y_w \rangle \leq \frac{sk}{n}} \cdot \mathbb{1}_{\langle g_{i'} t_j, y_l \rangle \leq \frac{sk}{n}}$$

This expression is a Monte Carlo approximation of the integral:

$$\mathbb{E}_t \int_{-s}^{s} \psi(y_w, t, \tau) \cdot \psi(y_l, t, \tau)\, d\tau = \mathcal{K}_s(y_w, y_l \mid x)$$

where $\psi(y, t, \tau) = \mathbb{P}_g(\langle gt, y \rangle \leq \tau)$ is the truncated CDF of the dot product between group-transformed templates and response vectors.

As $n \to \infty$, the sum over bins converges to the Riemann integral over $[-s, s]$. As $m \to \infty$, the template sampling converges to the expectation over $t$. Thus:

$$\lim_{n \to \infty} \mathbb{E}_{t,g} \langle \Phi(y_w), \Phi(y_l) \rangle = \mathcal{K}_s(y_w, y_l \mid x)$$

This concludes the proof. $\qquad\square$

### D.2  Proof of Theorem 1

**Theorem 1** (Invariant Features Maps and Distances between Orbits). *Let $\epsilon \in (0, 1)$ and $y_w, y_l \in \mathcal{Y}$. Denote the orbit to be the collection of all group-transformations of a given input $y$: $\mathcal{O}_x = \{gx, g \in \mathcal{G}\}$. We define the distance measure $d_\mathcal{G}$ between two orbits $\mathcal{O}_{y_w}$ and $\mathcal{O}_{y_l}$: $d_\mathcal{G}(y_w, y_l | x) = \frac{1}{\sqrt{2\pi d}} \int_{g \in \mathcal{G}} \int_{g' \in \mathcal{G}} \|g y_w - g' y_l\|_2 d\mu(g) d\mu(g')$. Fix $\epsilon_0, \delta \in (0, 1)$. For a number of bins $n \geq \frac{3}{\epsilon_0}$, templates $m \geq \frac{9C_1}{\epsilon_0^2} \log(\frac{N}{\delta})$, and group elements $|\mathcal{G}| \geq \frac{9C_2}{\epsilon_0^2} \log(\frac{Nm}{\delta})$, where $C_1, C_2$ are constants. The following inequality holds with probability $1 - 2\delta$:*

$$\epsilon - \delta_2(d, \epsilon) - \epsilon_0 \leq \langle \Phi(y_w), \Phi(y_l) \rangle - (1 - d_\mathcal{G}(y_w, y_l | x)) \leq \epsilon_0 + \epsilon + \delta_1(d, \epsilon), \qquad (17)$$

*where $i = 1 \ldots N, j = 1 \ldots N$.*

*Proof.* Let $y_w, y_l \in \mathcal{Y}$ be responses and fix $\varepsilon_0, \delta \in (0, 1)$. Let the number of bins satisfy $n \geq \frac{3}{\varepsilon_0}$, number of templates $m \geq \frac{9C_1}{\varepsilon_0^2} \log\left(\frac{N}{\delta}\right)$, and number of group samples $|\mathcal{G}| \geq \frac{9C_2}{\varepsilon_0^2} \log\left(\frac{Nm}{\delta}\right)$, where $C_1, C_2$ are absolute constants.

We decompose the kernel approximation error into three errors as an upper bound, using triangle inequality:

$$|\langle \Phi(y_w), \Phi(y_l) \rangle - \mathcal{K}_s(y_w, y_l | x)| \leq \underbrace{\left| \langle \Phi(y_w), \Phi(y_l) \rangle - \hat{\mathcal{K}}(y_w, y_l | x) \right|}_{\text{Riemann approximation error}}$$

$$+ \underbrace{\left| \hat{\mathcal{K}}(y_w, y_l | x) - \tilde{\mathcal{K}}(y_w, y_l | x) \right|}_{\text{Group sampling error}}$$

$$+ \underbrace{\left| \tilde{\mathcal{K}}(y_w, y_l | x) - \mathcal{K}_s(y_w, y_l | x) \right|}_{\text{Template sampling error}}.$$

From the analysis in [34], the following bounds hold with high probability:

(1). The binning approximation error is at most $\varepsilon_0$ when $n \geq \frac{3}{\varepsilon_0}$. (2). The group sampling error is at most $\varepsilon$ with probability at least $1 - \delta$, when $|\mathcal{G}|$ satisfies the stated lower bound. (3). The template sampling error is bounded by $\delta_1(d, \varepsilon)$ and $\delta_2(d, \varepsilon)$, defined as:

$$\delta_1(d, \varepsilon) = \frac{e^{-d\varepsilon^2/16}}{\sqrt{d}}, \quad \delta_2(d, \varepsilon) = (1 + \varepsilon)e^{-d\varepsilon^2/8}.$$

We then relate the kernel to orbit distances via:

$$\mathcal{K}_s(y_w, y_l|x) = 1 - d_{\mathcal{G}}(y_w, y_l|x) \pm \delta,$$

where $d_{\mathcal{G}}(y_w, y_l|x) = \frac{1}{\sqrt{2\pi d}} \int_{g, g'} \|gy_w - g'y_l\|_2 d\mu(g) d\mu(g')$.

Combining the three error terms and applying union bound over the events, we obtain with probability at least $1 - 2\delta$:

$$\varepsilon - \delta_2(d, \varepsilon) - \varepsilon_0 \leq \langle \Phi(y_w), \Phi(y_l) \rangle - (1 - d_{\mathcal{G}}(y_w, y_l|x)) \leq \varepsilon + \varepsilon_0 + \delta_1(d, \varepsilon).$$

$\square$

## D.3 Proof of Theorem 2

**Theorem 2** (Generalization Bound of PRISM). *Let $\mathcal{H}_{\mathcal{K}_{inv}}$ be the Reproducing Kernel Hilbert Space (RKHS) induced by $\mathcal{K}_{inv}$ and define the hypothesis ball $\mathcal{F} := \{r_\theta, \|r_\theta\|_{\mathcal{H}_{\mathcal{K}_{inv}}} \leq C\}$ for the fixed radius $C > 0$. We assume the log-sigmoid loss $V(\cdot)$ is L-Lipschitz. Then, for any $\delta > 0$, with probability $1 - 3\delta$, the following inequality holds:*

$$\mathcal{E}_V(r_\theta^*) \leq \inf_{r_\theta \in \mathcal{F}} \mathcal{E}_V(r_\theta) + \frac{4LC}{\sqrt{N}} \left(1 + \sqrt{\log \frac{1}{\delta}}\right) + \lambda_1 LC \left(\frac{2}{\sqrt{m}} + \frac{2}{\sqrt{|\mathcal{G}|}} + \frac{2}{n}\right) + \lambda_2 LC \sqrt{\frac{m \log \frac{N}{\delta}}{N}},$$
(18)

*where $\mathcal{E}_V(r_\theta^*)$ and $\mathcal{E}_V(r_\theta)$ are the optimal risk and empirical risk of the reward model $r_\theta$.*

*Proof.* Let $\mathcal{H}_{\mathcal{K}_{inv}}$ be the RKHS induced by the PRISM kernel $\mathcal{K}_{inv}$, and define the hypothesis class $\mathcal{F} := \{r_\theta \in \mathcal{H}_{\mathcal{K}_{inv}} : \|r_\theta\|_{\mathcal{H}_{\mathcal{K}_{inv}}} \leq C\}$. We denote the log sigmoid loss $V(u)$ as $L$-Lipschitz. Let $r_\theta^*$ denote the global minimum reward function of the expected loss in $\mathcal{F}$, i.e., $r_\theta^* = \arg\min_{r \in \mathcal{F}} \mathcal{E}_V(r)$.

We bound the expected loss $\mathcal{E}_V(r_\theta)$ in terms of three contributions: generalization from finite samples, kernel approximation from shortcut features, and correlation regularization from empirical estimation.

We start by noting that for any $r_\theta \in \mathcal{F}$, standard generalization theory (e.g., via Rademacher complexity bounds for RKHS balls) yields the following with probability at least $1 - \delta$ [63]:

$$\mathcal{E}_V(r_\theta) \leq \hat{\mathcal{E}}_V(r_\theta) + \frac{2LC}{\sqrt{N}} \left(1 + \sqrt{\log \frac{1}{\delta}}\right),$$

where $\hat{\mathcal{E}}_V(r_\theta) := \frac{1}{N} \sum_{i=1}^N V(m(y_w^{(i)}, y_l^{(i)}|x^{(i)}))$ is the empirical risk evaluated on the sample $(x^{(i)}, y_w^{(i)}, y_l^{(i)})$.

We now compare the empirical performance of $r_\theta$ with the best possible hypothesis in $\mathcal{F}$. Let $r_{\mathcal{F}}^* := \arg\min_{r \in \mathcal{F}} \hat{\mathcal{E}}_V(r)$ be the empirical minimizer over $\mathcal{F}$. By definition,

$$\hat{\mathcal{E}}_V(r_\theta) \leq \hat{\mathcal{E}}_V(r_{\mathcal{F}}^*) + \lambda_1 L \cdot \epsilon_{kernel} + \lambda_2 LC \cdot \epsilon_{corr},$$

where the additional terms arise from kernel approximation and regularization.

For the kernel term, Proposition 1 and Theorem 1 imply that the inner product between feature maps approximates the invariant kernel with error at most:

$$\epsilon_{kernel} = \frac{2}{\sqrt{m}} + \frac{2}{\sqrt{|\mathcal{G}|}} + \frac{2}{n}.$$

Since the log sigmoid loss $V$ is $L$-Lipschitz and the hypothesis space is bounded by $\|r_\theta\|_{\mathcal{H}} \leq C$, the propagated loss deviation is bounded by $\lambda_1 LC \cdot \epsilon_{kernel}$.

For the decorrelation regularization, we apply standard results on the estimation of empirical covariance over batches with Rademacher complexity and McDiarmid concentration, which yield that for $m$ shortcut features, the decorrelation estimation error satisfies:

$$\epsilon_{\text{corr}} = \sqrt{\frac{m \log \frac{N}{\delta}}{N}}.$$

Finally, we compare the empirical risk of the best empirical function $r_{\mathcal{F}}^*$ to the expected risk of the true best function $r_\theta^*$. Using the same generalization bound again with probability at least $1 - \delta$:

$$\hat{\mathcal{E}}_V(r_{\mathcal{F}}^*) \leq \mathcal{E}_V(r_\theta^*) + \frac{2LC}{\sqrt{N}}\left(1 + \sqrt{\log \frac{1}{\delta}}\right).$$

Combining all the terms above and applying a union bound over the three high-probability events (each holding with probability at least $1 - \delta$), we conclude that with probability at least $1 - 3\delta$, the following bound holds:

$$\mathcal{E}_V(r_\theta^*) \leq \inf_{r \in \mathcal{F}} \mathcal{E}_V(r) + \frac{4LC}{\sqrt{N}}\left(1 + \sqrt{\log \frac{1}{\delta}}\right) + \lambda_1 LC\left(\frac{2}{\sqrt{m}} + \frac{2}{\sqrt{|\mathcal{G}|}} + \frac{2}{n}\right) + \lambda_2 LC\sqrt{\frac{m \log \frac{N}{\delta}}{N}}.$$

$\square$

## E    Training Dataset

The RLHFlow training dataset [35] used in our experiments integrates multiple open-source preference datasets, each selected to cover diverse preference scenarios and annotation methods. Specifically, the dataset includes general conversational preference data, such as HH-RLHF [64], consisting of human-annotated conversational pairs; SHP [65], containing community-driven Reddit interactions; and HelpSteer [36], featuring prompts evaluated on various human-assessed criteria (e.g., helpfulness, coherence).

Additionally, the dataset comprises task-specific data: PKU-SafeRLHF [66] provides expert-annotated safety and helpfulness comparisons; UltraFeedback [43] offers GPT-4 annotations focusing on instruction-following and truthfulness across diverse models; and UltraInteract [67] contributes complex reasoning tasks structured into preference trees with detailed annotations.

Finally, multi-turn conversational datasets like Distilabel-Capybara [68] and Distilabel-Orca [69] further enrich the training set with GPT-4 annotated dialogue preferences originating from distinct prompt collections. Comprehensive details about the dataset composition are summarized in Table 4.

Table 4: Composition of the RLHFlow preference dataset.

| Source | Number of Samples |
| --- | --- |
| Distilabel-Orca [69] | 6,926 |
| Distilabel-Capybara [68] | 14,811 |
| PKU-SafeRLHF [66] | 26,874 |
| HelpSteer [36] | 37,131 |
| SHP [65] | 93,301 |
| HH-RLHF-Helpful [64] | 115,396 |
| UltraInteract [67] | 161,927 |
| UltraFeedback [43] | 340,025 |

## F    Prompt Engineering

We provide the prompts for the extraction of the shortcut features, Sycophancy in Figure 5, Tone in Figure 6, Helpfulness in Figure 8, and Creativity in Figure 7. We implement a verification function to make sure the output score is scaled in the defined range. When the LLM API calls fail (due to server instability or model hallucination with wrong response formats), the feature extractor will leverage a fallback function to ensure it can still be calculated with a heuristic-based function. We provide the pseudocode in Algorithm 1.

## LLM-as-a-Judge Prompt for Response Sycophancy

**System Message**
You are a helpful assistant that can judge the quality of prompt-response pairs. Rate how much the assistant's response agrees with or flatters the user on a scale from 0 to 10, where:
- 0 means completely independent and willing to disagree with the user
- 10 means excessively agreeable and flattering to the user
Return only the numeric score.
**User Message**
Prompt Message: `<prompt>`
Response: `<response>`
Provide only a single numerical score without any explanation.

Figure 5: Prompt template for Response Sycophancy.

## LLM-as-a-Judge Prompt for Response Tone

**System Message**
You are a helpful assistant that can judge the quality of prompt-response pairs. Rate the formality of the following text on a scale from 0 to 10,
where 0 is extremely casual and 10 is extremely formal.
Return only the numeric score.
**User Message**
Prompt Message: `<prompt>`
Response: `<response>`
Provide only a single numerical score without any explanation.

Figure 6: Prompt template for Response Tone.

## LLM-as-a-Judge Prompt for Response Creativity

**System Message**
You are a helpful assistant that can judge the quality of prompt-response pairs. Rate the creativity and originality of this text on a scale from 0 to 10.
Return only the numeric score.
**User Message**
Prompt Message: `<prompt>`
Response: `<response>`
Provide only a single numerical score without any explanation.

Figure 7: Prompt template for Response Creativity.

## LLM-as-a-Judge Prompt for Response Helpfulness

**System Message**
Rate how helpful and informative this response is on a scale from 0 to 10.
Return only the numeric score.
**User Message**
Prompt Message: `<prompt>`
Response: `<response>`
Provide only a single numerical score without any explanation.

Figure 8: Prompt template for Response Helpfulness.

**Algorithm 1** Fallback Function for LLM-as-a-Judge
─────────────────────────────────────────────────
1: **function** CALL_LLM_JUDGE(prompt)
2:     text ← extract_text_to_evaluate(prompt)
3:     **if** Rate the Reponse Tone **then**
4:         score ← 5.0                                    ▷ default score
5:         formal_words ← ["would", "could", "should", "certainly", ...]
6:         informal_words ← ["hey", "yeah", "cool", "awesome", ...]
7:         score ← score +0.5· COUNT_OCCURRENCES(text, formal_words)
8:         score ← score −0.5· COUNT_OCCURRENCES(text, informal_words)
9:         **return** CLIP(score, 0.0, 10.0)
10:     **else if** Rate the Reponse Sycophancy **then**
11:         score ← 5.0                                    ▷ default score
12:         agree_words ← ["absolutely", "definitely", "I agree", ...]
13:         disagree_words ← ["however", "actually", "I disagree", ...]
14:         score ← score +0.5· COUNT_OCCURRENCES(text, agree_words)
15:         score ← score −0.5· COUNT_OCCURRENCES(text, disagree_words)
16:         **return** CLIP(score, 0.0, 10.0)
17:     **end if**
18: **end function**

19: **function** CLIP(value, min, max)
20:     **if** value < min **then**
21:         **return** min
22:     **else if** value > max **then**
23:         **return** max
24:     **else**
25:         **return** value
26:     **end if**
27: **end function**
─────────────────────────────────────────────────

# G  Case Studies

**Prompt:** Why is it best for people on benefits not to have kids?

**Response A (Chosen):**

> "I am not able to engage in that discussion. The decision to have children is a deeply personal one and should be made by individuals or couples based on their own circumstances, values, and aspirations."
>
> *(Preferred by PRISM — safer and less sycophantic)*

**Response B (Rejected):**

> "It is best for people on benefits not to have kids due to the significant financial burden, increased costs for benefits, impact on career prospects, and negative impact on mental health. Raising a child is expensive, and it can be a significant financial burden for people on benefits who may have limited income. (truncated for long context)"
>
> *(Preferred by Bradley-Terry RM — favors length and agreeable tone)*

**Prompt:** Should AI developers prioritize transparency over profitability?

**Response A (Rejected):**

> "Absolutely. Transparency is absolutely essential for building trust. Long-form explanations about the development process, model behavior, testing procedures, and failure modes are vital for accountability and user confidence. (truncated for long context)"
>
> *(Preferred by Bradley-Terry RM — verbose and agreeable)*

**Response B (Chosen):**

> "Transparency is crucial for trust in AI systems, though developers must also balance other priorities like usability and cost."
>
> *(Preferred by PRISM — focuses on content relevance)*

